# Alcoholic Liver Disease-Related Hepatocellular Carcinoma: Characteristics and Comparison to General Slovak Hepatocellular Cancer Population

Dominik Šafčák [1,2], Sylvia Dražilová [2,3], Jakub Gazda [2,3], Igor Andrašina [1], Svetlana Adamcová-Selčanová [4], Radovan Barila [5], Michal Mego [6], Marek Rác [7], Ľubomír Skladaný [4], Miroslav Žigrai [8], Martin Janičko [2,3,*] and Peter Jarčuška [2,3]

[1] Department of Radiotherapy and Oncology, East Slovakia Institute of Oncology, 04191 Košice, Slovakia
[2] 2nd Department of Internal Medicine, Louis Pasteur University Hospital, 04011 Košice, Slovakia
[3] Department of Internal Medicine, Faculty of Medicine, Pavol Jozef Safarik University in Kosice, 04011 Košice, Slovakia
[4] Department of Internal Medicine, F.D. Roosevelt University Hospital, 97517 Banská Bystrica, Slovakia
[5] Oncological Cluster, Saint Michael Hospital Michalovce, 07101 Michalovce, Slovakia
[6] Department of Clinical Oncology, National Oncology Institute of Slovakia, 83310 Bratislava, Slovakia
[7] Department of Internal Medicine, University Hospital Nitra, 94901 Nitra, Slovakia
[8] Department of Internal Medicine, University Hospital in Bratislava, 83101 Bratislava, Slovakia
* Correspondence: martin.janicko@upjs.sk; Tel.: +421-55-234-3437

**Abstract:** Hepatocellular carcinoma (HCC) has multiple molecular classes that are associated with distinct etiologies and, besides particular molecular characteristics, that also differ in clinical aspects. We aim to characterize the clinical aspects of alcoholic liver disease-related HCC by a retrospective observational study that included all consequent patients diagnosed with MRI or histologically verified HCC in participating centers from 2010 to 2016. A total of 429 patients were included in the analysis, of which 412 patients (96%) had cirrhosis at the time of diagnosis. The most common etiologies were alcoholic liver disease (ALD) (48.3%), chronic hepatitis C (14.9%), NAFLD (12.6%), and chronic hepatitis B (10%). Patients with ALD-related HCC were more commonly males, more commonly had cirrhosis that was in more advanced stages, and had poorer performance status. Despite these results, no differences were observed in the overall (median 8.1 vs. 8.5 months) and progression-free survival (median 4.9 vs. 5.7 months). ALD-HCC patients within BCLC stage 0–A less frequently received potentially curative treatment as compared to the control HCC patients (62.2% vs. 87.5%, $p = 0.017$); and in patients with ALD-HCC liver function (MELD score) seemed to have a stronger influence on the prognosis compared to the control group HCC. Systemic inflammatory indexes were strongly associated with survival in the whole cohort. In conclusion, alcoholic liver disease is the most common cause of hepatocellular carcinoma in Slovakia, accounting for almost 50% of cases; and patients with ALD-related HCC more commonly had cirrhosis that was in more advanced stages and had poorer performance status, although no difference in survival between ALD-related and other etiology-related HCC was observed.

**Keywords:** liver cirrhosis; cancer; alcohol

## 1. Introduction

Alcohol-related liver disease (ALD) is one the most prevalent liver diseases worldwide. The latest WHO Global status report on alcohol and health estimates approximately three million alcohol-contributed deaths in 2016, which represents 5.3% of all deaths worldwide. Alcohol was generally responsible for 7.2% of premature deaths worldwide. Chronic alcohol abuse over years results in alcoholic fatty liver in most patients. Approximately one third of these subjects will progress into alcoholic steatohepatitis, and up to 20% will

develop alcohol cirrhosis, which may progress to hepatocellular carcinoma for 0.5–2.6% annually [1].

Liver cancer is the sixth most frequently diagnosed cancer and the third most common cause of cancer-related deaths worldwide in 2020. Among all histological subtypes, hepatocellular carcinoma (HCC) represents 75%–85% of all diagnosed cases. Almost 90% of cases develop in the context of underlying chronic liver disease [2].

Genomic studies that evaluated the whole HCC genome using high-throughput methods have identified at least two distinctive mutation patterns called molecular classes. Proliferative class is characterized by activation of RAS, mTOR and insulin-like growth factor. Non-proliferative class displays mutations in beta-catenin gene (CTNNB1) [3] and is more closely associated with chronic inflammation. These classes are usually associated with different etiologies, although the clinical differences have not been clearly attributed to different molecular patterns. Alcoholic liver disease-related HCC is usually associated with the non-proliferative class of HCC [2]. Therefore, one of the hypotheses was that ALD-related HCC would be more commonly diagnosed in cirrhotic patients, associated with the degree of liver dysfunction, and that patients with ALD-related HCC (ALD-HCC) would have better outcomes than patients with HCC due to other etiologies. Chronic alcohol intake also influences the immune system of the tumor microenvironment. Increased gut permeability enhances the translocation of immunomodulatory microbiota-derived pathogen-associated molecular patterns (PAMPs), which suppress hepatic immune responses. Presence of neutrophils in liver parenchyma is typical for alcoholic hepatitis, and alcoholic steatohepatitis is associated with increased accumulation of MSDCs and suppression of T-cell recruitment. Changes in the immune system response are also presented in the peripheral blood. Many studies show the negative impact of increased inflammatory indexes such as neutrophil to lymphocyte ratio (NLR) [4], platelet to lymphocyte ratio (PLR) [5], and systemic immune inflammation index (SII) [6] on the overall survival of patients with HCC.

The aims of this study are (a) to describe the differences between patients with ALD-related HCC and patients with HCC in chronic liver disease (CLD) caused by other etiologies, and (b) to describe the influence of systemic inflammation on the survival of this cohort of patients.

## 2. Methods

We performed a multicenter retrospective study of patients diagnosed and treated for HCC at eight centers in Slovakia during the period from 2010 to 2016. All relevant patients were screened for eligibility according to the inclusion criteria.

### 2.1. Patient Selection

The inclusion criterium was the diagnosis of HCC consistent with EASL-EORTC guidelines [7] (HCC confirmed by either histopathological examination or magnetic resonance imaging). The exclusion criteria were as follows: (a) etiology not available, (b) cryptogenic etiology, and (c) combined or rare etiologies. Patients were managed according to local standards valid at the time of diagnosis.

### 2.2. Data Collection

All data were collected retrospectively from the patients' charts. Case report forms were completed by D.S. with the on-call assistance of P.J. Collected data included baseline blood test results at the time of diagnosis (hematology, biochemistry and hemocoagulation). If any condition that might have influenced baseline values was present (e.g., acute bacterial infection), the laboratory results from a later time were used. Data about the underlying liver disease, tumor characteristics, liver function, performance status, comorbidities, treatment, and outcomes were collected. Liver cirrhosis was diagnosed either by histopathological examination of the resected/explanted liver parenchyma or by a combination of clinical imaging (ultrasonography, CT, magnetic resonance imaging and

laboratory findings). The Child–Pugh score was used to estimate cirrhosis severity. The performance status was evaluated using the Eastern Cooperative Oncology Group (ECOG) Scale. The CT scans for thorax, abdomen and pelvis minor were used to identify extrahepatic spread and for final staging. Barcelona Clinic Liver Cancer (BCLC) staging system was used to assess prognosis. We also calculated the neutrophil to lymphocyte ratio (NLR), the platelet to lymphocyte ratio (PLR), the serum aspartate aminotransferase to platelet ratio index (APRI).

### 2.3. Inflammatory Indexes Were Calculated as:

(a) Systemic inflammation index (SII): SII = P $\times$ N/L, where P, N and L are the peripheral blood platelet, neutrophil and lymphocyte counts per liter, respectively. The optimum cut-off point for SII for a favorable prognosis was determined to be $\geq$330 $\times$ 109 cells/L for adverse prognosis [6].

(b) Neutrophil–lymphocyte ratio (NLR): N/L where N and L are the peripheral neutrophil and lymphocyte counts per liter, respectively, and the cut-off used was >4 for adverse prognosis [4].

(c) Thrombocyte lymphocyte ratio (TLR): T/L where T and L are the peripheral blood platelet and lymphocyte counts per liter, respectively, and the cut-off used was >150 for adverse prognosis [5].

Alcohol consumption thresholds of 20 g per day in women and 30 g per day in men, confirmed in medical records, were used to determine the high risk of alcoholic liver disease.

The treatment response was evaluated using CT scans or MR imaging according to the modified Response Evaluation Criteria in Solid Tumor. Finally, CRF also included the date of death extracted either from the patients' medical records or from the database of Slovak Health Care Surveillance Authority.

The study protocol is in accordance with the 1964 Helsinki declaration, its later amendments, and the principles of good clinical practice. The study protocol was approved by The Ethics Committee of East Slovakia Oncological Institute, on 27 May 2021 (approval code, EK/2/05/2021). The committee waived the need for the specific patients' informed consent due to the retrospective nature of the data collection (the data already existed and were not a result of a research activity) and the usage of anonymous data only.

### 2.4. Statistical Analysis

Patients were classified according to liver disease etiology to the ALD-related HCC group and other etiologies (composite control group). Patients were censored per analysis based on the availability of data (retrospective study). Survival is reported in months as median (min–max). Variables are reported as absolute and relative counts (categorical) or mean $\pm$ standard error of mean (interval variables). Inflammatory indexes and AFP are reported as median (interquartile range) because of skewed distribution. Inflammatory indices are calculated as referenced here and were used after natural logarithm transformation due to extremely skewed distribution. Ln transformation was also necessary for AFP levels. Baseline comparison of categorical variables—chi-squared, continuous variables T-test or Mann–Whitney test, respecting the tests' assumptions. Survival curves were constructed using the Kaplan–Meier procedure with log-rank comparison of factors. Adjusted survival hazard ratios produced by Cox regression.

### 3. Results

Overall, 483 HCC patients were screened: 54 patients were excluded based on missing data about etiology of chronic liver disease, and 429 patients were included in the analysis. A total of 412 patients (96%) had cirrhosis at the time of diagnosis. The most common etiology (Table 1) was alcoholic liver disease alone (207 pts; 48.3%), or in combination with chronic hepatitis B or C (9 pts; 2.1%), followed by chronic hepatitis C (64 pts; 14.9%), NAFLD (54 pts; 12.6%), chronic hepatitis B (43 pts; 10%), with one hepatitis B and C

coinfected patient, and cryptogenic etiology (40pts; 9.3%). Eleven patients (2.6%) had liver disease due to various minor etiologies. Statistical significance for differences in counts $p < 0.0001$ (chi-squared). Median follow up was 252 days (1–4725 days). Figure 1 shows STROBE flowchart of the data collection and analysis.

**Table 1.** Etiology of chronic liver disease.

| Etiology | N | % | 95% CI |
|---|---|---|---|
| Alcoholic liver disease | 207 | 48.3 | 43–53 |
| Chronic hepatitis C | 64 | 14.9 | 11.7–18.7 |
| NAFLD | 54 | 12.6 | 9.6–16.1 |
| Chronic hepatitis B | 43 | 10 | 7.4–13.3 |
| Alcohol in combination | 9 | 2.1 | 0.9–3.9 |
| HBV + HCV coinfection | 1 | 0.2 | 0.01–1.3 |
| Cryptogenic liver disease | 40 | 9.3 | 6.7–12.5 |

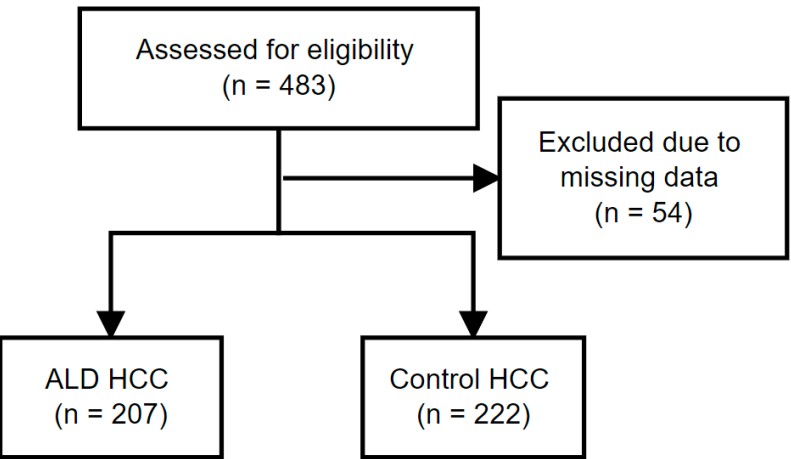

**Figure 1.** Flowchart of the study analysis according to STROBE guidelines.

Baseline comparison (Table 2) showed that patients with ALD-related HCC were more commonly males, more commonly had cirrhosis that was in more advanced stages, and had poorer performance status. Despite these results, no differences were observed in overall and progression-free survival.

**Table 2.** Baseline characteristics of the study cohort.

| Parameter | n | ALD | n | Control Group | P |
|---|---|---|---|---|---|
| Patients | 207 | 207 (48.3%) | 222 | 222 (51.7%) | N/A |
| Male sex | 207 | 153(73.9%) | 222 | 127(57.2%) | <0.0001 |
| Age at diagnosis (years) | 207 | 66.6 ± 0.5 | 222 | 65.0 ± 0.8 | 0.112 |
| Diabetes mellitus present | 207 | 72 (35%) | 220 | 86(39%) | 0.377 |
| Statin use at baseline | 207 | 10(4.8%) | 222 | 15(6.8%) | 0.395 |
| BCLC stage 0–A | | 37(17.9%) | | 33(14.9%) | |
| BCLC stage B | 207 | 56(27.1%) | 221 | 70(31.5%) | 0.079 |
| BCLC stage C | | 62(30.0%) | | 83(37.4%) | |
| BCLC stage D | | 52(25.1%) | | 36(16.2%) | |
| Cirrhosis at diagnosis | 207 | 206(99.5%) | 221 | 206(93.2%) | <0.0001 |
| Child–Pugh points | 196 | 7.4 ± 0.14 | 209 | 6.7 ± 0.13 | <0.0001 |

**Table 2.** *Cont.*

| Parameter | n | ALD | n | Control Group | P |
|---|---|---|---|---|---|
| MELD score | 196 | $6.5 \pm 0.41$ | 213 | $6.2 \pm 0.44$ | <0.0001 |
| Performance status ECOG 0 | | 2(1%) | | 8(3.6%) | |
| Performance status ECOG 1 | 207 | 151(72.9%) | 222 | 182(82%) | 0.009 |
| Performance status ECOG 2 | | 52(25.1%) | | 31(14.0%) | |
| Performance status ECOG 3 | | 2(1.0%) | | 1(0.5%) | |
| BMI (kg/m$^2$) | 37 | $28.69 \pm 0.57$ | 42 | $29.62 \pm 0.71$ | 0.320 |
| CRP (mg/L) | 167 | $27.62 \pm 2.35$ | 174 | $25.69 \pm 2.91$ | 0.608 |
| AST (ukat/L) | 207 | $1.51 \pm 0.1$ | 221 | $2.58 \pm 0.81$ | 0.203 |
| ALT (ukat/L) | 207 | $0.88 \pm 0.07$ | 221 | $1.08 \pm 0.06$ | 0.031 |
| ALP (ukat/L) | 205 | $3.27 \pm 0.21$ | 219 | $3.3 \pm 0.19$ | 0.935 |
| GGT (ukat/L) | 207 | $4.07 \pm 0.28$ | 220 | $3.58 \pm 0.25$ | 0.194 |
| Total bilirubin (umol/L) | 207 | $44.31 \pm 4.44$ | 221 | $31.89 \pm 2.72$ | 0.016 |
| Direct bilirubin (umol/L) | 166 | $24.52 \pm 3.47$ | 158 | $18.67 \pm 2.19$ | 0.159 |
| Albumin (g/L) | 203 | $32.49 \pm 0.53$ | 216 | $33.64 \pm 0.55$ | 0.131 |
| INR | 199 | $1.27 \pm 0.01$ | 215 | $1.26 \pm 0.02$ | 0.873 |
| Neutrophiles ($\times 10^9$/L) | 181 | $4.99 \pm 0.26$ | 206 | $4.74 \pm 0.24$ | 0.466 |
| Lymphocytes ($\times 10^9$/L) | 152 | $1.28 \pm 0.06$ | 176 | $1.36 \pm 0.05$ | 0.316 |
| Platelets ($\times 10^9$/L) | 207 | $164.36 \pm 7.59$ | 219 | $192.93 \pm 8.61$ | 0.014 |
| Cholesterol (mmol/L) | 76 | $13.19 \pm 8.81$ | 98 | $4.15 \pm 0.17$ | 0.245 |
| HDL (mmol/L) | 53 | $0.99 \pm 0.08$ | 75 | $1.37 \pm 0.19$ | 0.104 |
| LDL (mmol/L) | 52 | $2.79 \pm 0.19$ | 73 | $3.38 \pm 0.94$ | 0.601 |
| Triglycerides (mmol/L) | 59 | $1.25 \pm 0.08$ | 93 | $1.26 \pm 0.09$ | 0.948 |
| Creatinine (μmol/L) | 204 | $98.96 \pm 4.2$ | 219 | $85.43 \pm 2.4$ | 0.005 |
| Urea (mmol/L) | 206 | $7.61 \pm 0.59$ | 219 | $6.91 \pm 0.56$ | 0.395 |
| Na (mmol/L) | 204 | $137.4 \pm 1.58$ | 218 | $137.6 \pm 0.36$ | 0.898 |
| AFP (kIU/L) | 180 | 28.1 (864.1) | 195 | 44 (666.7) | 0.094 |
| NLR | 150 | 4 (3.51) | 174 | 3.42 (3.1) | 0.056 |
| TLR | 151 | 129.8 (136.7) | 175 | 141.2 (127.7) | 0.803 |
| APRI | 207 | $2 \pm 0.15$ | 219 | $3.01 \pm 0.82$ | 0.240 |
| SII ($10^9$ cells/L) | 150 | 563 (973.3) | 174 | 552.9 (972.2) | 0.773 |
| No. lesions | 192 | 2 (5) | 212 | 2 (4,8) | 0.629 |
| Max lesion size (mm) | 192 | $59.8 \pm 3.4$ | 212 | $63.7 \pm 2.8$ | 0.053 |
| MAVI—0 | | 168(81.2%) | | 169(76.1%) | |
| MAVI—1 | 207 | 16(7.7%) | 222 | 31(14.0%) | 0.117 |
| MAVI—2 | | 23(11.1%) | | 22(9.9%) | |
| MIVI—0 | | 163(78.7%) | | 163(73.4%) | |
| MIVI—1 | 207 | 41(19.8%) | 222 | 55(24.8%) | 0.436 |
| MIVI—2 | | 3(1.4%) | | 4(1.8%) | |

**Table 2.** *Cont.*

| Parameter | n | ALD | n | Control Group | P |
|---|---|---|---|---|---|
| DFS/PFS (months) | 207 | 4.9(0–154.9) | 222 | 5.7(0.1–114.7) | 0.528 |
| OS (months) | 207 | 8.1(0–154.9) | 222 | 8.5(0.1–122.9) | 0.315 |

BCLC—Barcelona clinic liver cancer, CRP—C-reactive protein, AFP—alpha fetoprotein, AST—aspartate amino-transferase, ALT—alanine aminotransferase, GGT—gamma glutamyltransferase, INR—international normalization ratio, NLR—neutrophile–lymphocyte ratio, TLR—thrombocyte-lymphocyte ratio, SII—systemic inflammatory index, MAVI—macrovascular invasion, MIVI—microvascular invasion, DFS—disease free survival, PFS—progression-free survival, OS—overall survival.

First administered treatment in the respective groups is summarized in Table 3. Treatment data was available for the whole cohort, however, in the statistical analysis we omitted patients who received second line chemotherapy (four patients; 0.8%) or a combination of TACE and sorafenib (one patient; 0.2%) as the first treatment option. There was no difference in the proportion of patients receiving potentially curative or palliative treatments between ALD-related HCC and control group (palliative treatment being the more common modality at 83.6% in ALD-related HCC vs 80.1% in the control group; *p* = 0.326) in the whole cohort. If analyzed by BCLC stages, ALD-HCC patients within BCLC stage 0–A less frequently received potentially curative treatment as compared to the control HCC patients (62.2% vs. 87.5%) Furthermore, 37.8% of patients with ALD-related HCC were offered only palliative treatment even in the BCLC 0-A stage compared to only 12.5% of patients from the control group HCC (*p* = 0.017).

**Table 3.** Treatment used in the first line.

| Treatment | ALD | | Control | |
|---|---|---|---|---|
| | n | % | n | % |
| Resection | 14 | 6.8% | 30 | 10.9% |
| RFA | 9 | 4.3% | 13 | 4.7% |
| TACE | 46 | 22.2% | 55 | 19.9% |
| Sorafenib | 49 | 23.7% | 93 | 33.7% |
| Supportive | 76 | 36.7% | 70 | 25.4% |
| Sorafenib + TACE | 1 | 0.5% | 0 | 0.0% |
| Second line chemo | 1 | 0.5% | 3 | 1.1% |
| LTx | 11 | 5.3% | 12 | 4.3% |

ALD—Alcoholic liver disease, RFA—radiofrequency ablation, TACE—transarterial chemoembolisation, LTx—liver transplant.

Treatment response assessment was available in 479 patients. No significant difference in first line therapy treatment response was observed between ALD-related HCC and the control group, *p* = 0.122 (Table 4).

**Table 4.** Treatment response.

| Response | ALD | | Control | |
|---|---|---|---|---|
| | n | % | n | % |
| Complete response | 36 | 17.60 | 56 | 20.4 |
| Partial response | 27 | 13.20 | 25 | 9.1 |
| Stable disease | 23 | 11.30 | 48 | 17.5 |
| Progressive disease | 118 | 57.80 | 146 | 53.1 |

No difference was observed between the ALD-related HCC group and the control group in unadjusted overall survival and unadjusted progression-free survival. Figure 2 shows Kaplan–Meier survival curves for both groups along with confidence intervals. There is significant overlap of confidence intervals during the complete follow-up period. Moreover, after adjustment for age, sex, first line treatment and BCLC class, there was no

difference in the hazard ratios between ALD-related HCC and the control group (HR 1.057, 95% CI 0.866–1.291; *p* = 0.584).

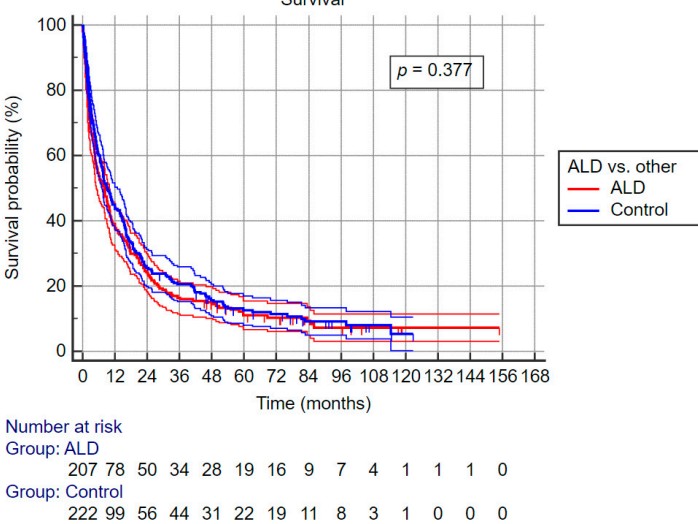

**Figure 2.** Overall survival according to study groups by Kaplan–Meier analysis.

The analysis of individual predictors of survival (Table 5) shows that significant predictors are the same for both ALD-related HCC and control group patients, except for age, which is not significantly related to survival in the control group HCC. However, the univariate HRs suggest that survival in ALD-related HCC is more dependent on liver function than in non-ALD-HCC. The presence of diabetes mellitus did not have an association with mortality. Use of statins was infrequent, thus not included in the analysis. To assess the relative weight of the survival predictors in both ALD-related HCC and control group HCC we input the predictors into a multivariate model that contained parameters reflecting liver function (MELD), performance status, and tumor parameters (max size and number of lesions). In this comparison (Table 6), in the case of ALD-related HCC MELD score seems to have a higher relative impact on survival than in control group HCC (adjusted MELD OR 1.65 vs 1.305 albeit with overlapping confidence intervals).

**Table 5.** Univariate logistic regression analysis of variables associated with survival.

| Variable | ALD-Related HCC | | Control Group HCC | |
|---|---|---|---|---|
| | **Cox HR** | **P** | **Cox HR (95% CI)** | **P** |
| Age | 1.030 (1.011–1.050) | 0.002 | 1.009 (0.999–1.020) | 0.087 |
| Sex | 1.262 (0.911–1.748) | 0.161 | 1.2 (0.932–1.545) | 0.156 |
| DM present | 1.125 (0.834–1.517) | 0.441 | 1.052 (0.789–1.401) | 0.731 |
| BCLC—0–A (ref) | | | | |
| BCLC—B | 1.751 (1.077–2.849) | <0.024 | 2.445 (1.496–3.994) | <0.0001 |
| BCLC—C | 4.884 (2.983–7.996) | <0.0001 | 7.733 (4.694–12.747) | <0.0001 |
| BCLC—D | 10.856 (6.485–18.173) | <0.0001 | 24.968 (14.173–43.983) | <0.0001 |
| CRP | 1.013 (1.008–1.017) | <0.0001 | 1.009 (1.006–1.012) | <0.0001 |
| PLR | 1.590 (1.211–2.087) | 0.001 | 1.760 (1.366–2.267) | <0.0001 |
| NLR | 2.037 (1.563–2.654) | <0.0001 | 2.015 (1.559–2.605) | <0.0001 |
| AFP | 1.134 (1.073–1.198) | <0.0001 | 1.073 (1.015–1.134) | 0.012 |
| SII | 1.736 (1.437–2.098) | <0.0001 | 1.662 (1.411–1.957) | <0.0001 |
| MELD | 2.076 (1.1545–2.788) | <0.0001 | 1.402 (1.119–1.758) | 0.003 |
| Child–Pugh | 1.222 (1.132–1.318) | <0.0001 | 1.139 (1.064–1.220) | <0.0001 |
| Max lesion size | 1.005 (1.002–1.009) | 0.002 | 1.006 (1.003–1.008) | <0.0001 |
| No. of lesions | 1.156 (1.104–1.202) | <0.0001 | 1.101 (1.072–1.12)1 | <0.0001 |

**Table 5.** *Cont.*

| Variable | ALD-Related HCC | | Control Group HCC | |
|---|---|---|---|---|
| | Cox HR | P | Cox HR (95% CI) | P |
| ECOG—0 (ref) | | <0.0001 | | <0.0001 |
| ECOG—1 | 0.784 (0.194–3.176) | 0.733 | 0.659 (0.405–1.070) | 0.092 |
| ECOG—2 | 3.563 (0.862–14.727 | 0.079 | 2.543 (1.453–4.448) | 0.001 |
| ECOG—3 | 5.352 (0.737–38.856) | 0.097 | 4.420 (1.006–19.412) | 0.049 |

ECOG—Eastern cooperative oncology group performance status, BCLC—Barcelona clinic liver cancer, CRP—C-reactive protein, AFP—alpha fetoprotein, NLR—neutrophile–lymphocyte ratio, TLR—thrombocyte-lymphocyte ratio, SII—systemic inflammatory index.

**Table 6.** Analysis of tumor characteristics, performance status and MELD in both studied groups.

| Variable | ALD-HCC | | | Control Group HCC | | |
|---|---|---|---|---|---|---|
| | Sig. | Exp (B) | 95% CI | Sig. | Exp (B) | 95% CI |
| Number of Tu | <0.0001 | 1.176 | 1.122–1.232 | <0.0001 | 1.117 | 1.07–1.166 |
| Max Tu size | <0.0001 | 1.007 | 1.003–1.011 | 0.001 | 1.007 | 1.003–1.011 |
| ECOG 0 | <0.0001 | | | <0.0001 | | |
| ECOG 1 | 0.392 | 0.540 | 0.132–2.209 | 0.084 | 0.508 | 0.236–1.095 |
| ECOG 2 | 0.233 | 2.417 | 0.566–10.314 | 0.182 | 1.852 | 0.749–4.582 |
| ECOG 3 | 0.189 | 3.945 | 0.509–30.588 | 0.132 | 5.160 | 0.611–43.585 |
| MELD | 0.003 | 1.650 | 1.18–2.306 | 0.050 | 1.305 | 1–1.705 |

ECOG—Eastern cooperative oncology group performance status.

Systemic inflammatory index, neutrophil–lymphocyte ratio and thrombocyte–lymphocyte ratio are relatively new indexes that are associated with cancer survival. In this cohort there was no difference in all three indexes between ALD-related HCC and control group HCC. Furthermore, all three indexes were significantly associated with survival. Therefore, we evaluated their association with survival for the whole cohort. Figure 3 shows that patients with SII of more than $330 \times 109$ cells/L have significantly better survival; the same was observed for NLR—Figure 4 (cut-off for favorable prognosis was ≤4) and also TLR—Figure 5 (cut-off for favorable prognosis was ≤150).

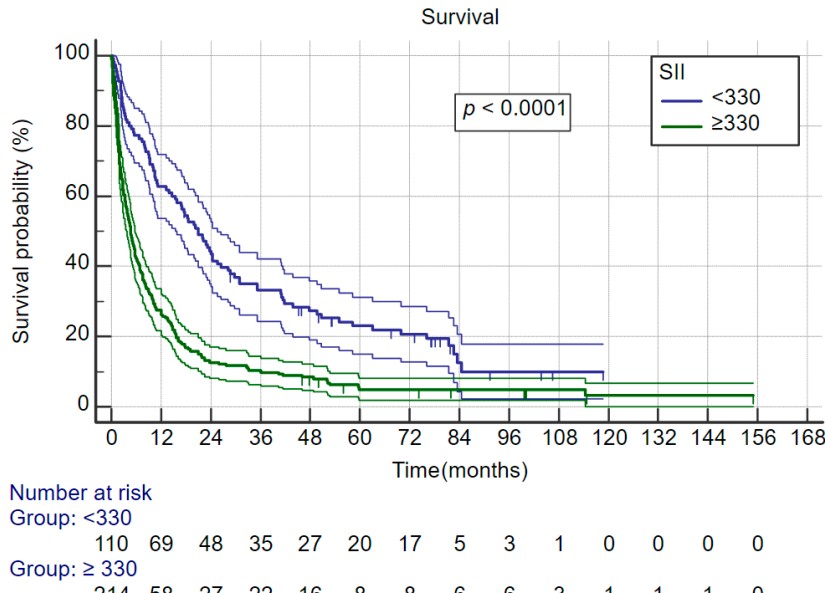

**Figure 3.** Kaplan–Meier survival curves for the whole cohort, split by systemic inflammatory index categories.

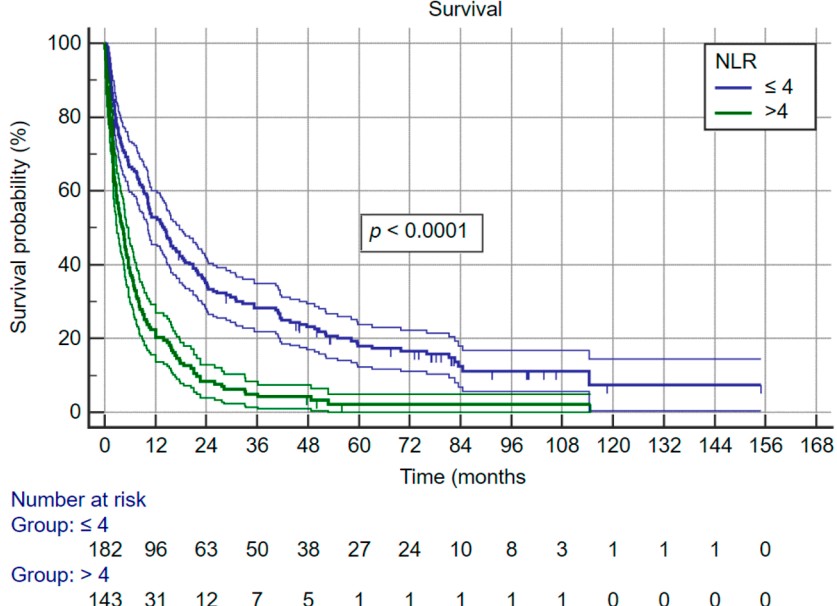

**Figure 4.** Kaplan–Meier survival curves for whole cohort, split by neutrophil–lymphocyte ratio categories.

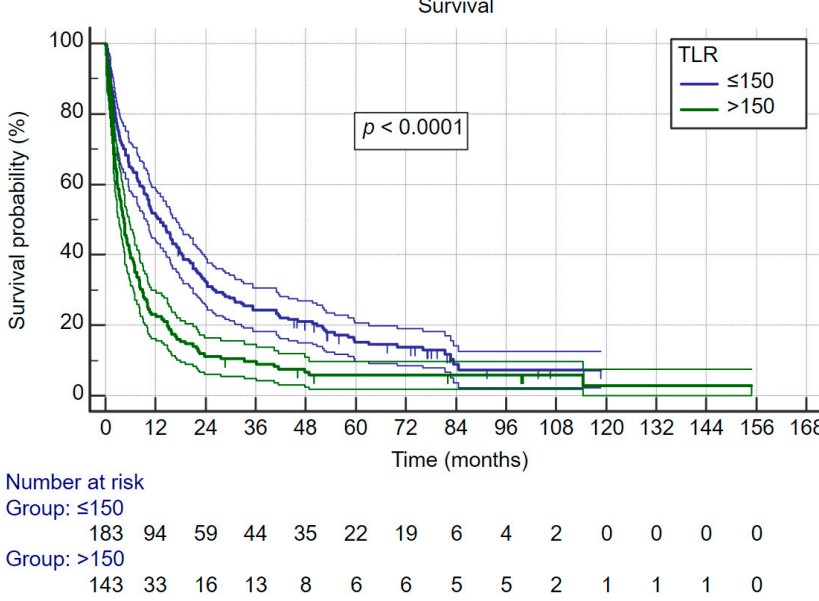

**Figure 5.** Kaplan–Meier survival curves for whole cohort, split by thrombocyte–lymphocyte ratio categories.

## 4. Discussion

The risk factors for HCC display significant geographical variability. Data from the Global HCC BRIDGE study show that the most common HCC risk factors are chronic hepatitic C in the US and chronic hepatitis B in China and South-East Asian countries. However, this study completely lacked data from Central and Eastern Europe [8]. The only other sources on HCC risk factors are expert estimates based on opinions and conjectures between alcohol consumption and cancer incidence rates. Particularly in Slovakia, no empirical data were available until now. These estimates show that alcoholic liver disease is one of the most serious risk factors for HCC. According to GLOBOCAN estimates, 245,000 cases of liver cancer out of a total of 854,000 cases worldwide were due to this disease in 2015. The percentage of ALD in newly diagnosed HCC worldwide ranges from 6% in Iran to 50–60% in Eastern European countries [9]. The annual incidence of HCC in patients with cirrhosis due to alcoholic liver disease ranges from 2.1–5.6% [9–12]. This risk can be further increased by other factors such as obesity [13] or smoking [14]. In this

study we tried to partially fill in the missing data on the HCC risk factors in Slovakia. Since the data were collected from centers throughout Slovakia, we consider these results as representative for our country. As there are many commonalities between the countries in Eastern Europe, particularly regarding alcohol consumption [15] and chronic viral hepatitis prevalence [16], these results could also be indicative of the HCC risk in the surrounding countries. It is clearly visible that, in this geographic area, alcoholic liver disease is the dominant risk factor of HCC with a wide margin, including accounting for confidence intervals. These results show that ALD was the major risk factor for HCC for 48.3% of patients. This value is higher compared to the estimates published by Baecker et al. using the GLOBOCAN 2012 data, who calculated the population attributable fraction risk of HCC for ALD at 36% for Central Europe [17], and is almost the same as the estimate reported by the Global Burden of Disease Liver Cancer Collaboration which estimated the alcoholic liver disease contribution of liver cancer deaths to be 46% in central Europe [18].

Patients in our cohort who had ALD-related HCC were more likely to have cirrhosis at the time of diagnosis compared with controls. The same results were reported in a retrospective analysis by Hester et al. [19], in which cirrhosis at the time of HCC diagnosis was observed more frequently in patients with ALD-related HCC (94.8%) compared to HBV (85.1%) and NASH-related HCC (83.7%). The prevalence of cirrhosis in HCC patients with HCV background was comparable to ALD (93.3%) [19]. Almost identical results were reported by a prospective study of 103 hepato-gastroenterology centers in France (93.9% of patients with ALD-related HCC vs. 73.2% for non-alcoholic etiology, $p < 0.001$) [20].

Alcoholic liver disease is usually included in the non-proliferative molecular class of HCC. Therefore, we expected to observe at least some clinical differences at the time of diagnosis. Indeed, as is shown in Table 2, patients with ALD-related HCC had on average marginally smaller maximal tumor size, which was borderline insignificant, and on the other hand these patients had poorer liver function as shown by worse Child–Pugh and MELD scores. A similar study by deLemos et al. from five clinical centers in the US on a much larger sample size ($n = 5327$) did not find a difference in the maximal tumor size; however, patients with ALD-HCC were less frequently diagnosed within the Milan criteria, at least numerically, although the difference approached significance [21]. Despite expectations, patients did not differ significantly in AFP levels, although numerical difference in medians was quite substantial. This was probably due to extreme variability of AFP levels in both ALD-HCC and control HCC groups. The previously referenced study by deLemos indeed found that AFP levels were lower in ALD-HCC despite identical maximal tumor size [21]. Poorer liver function is also commonly associated with ALD-HCC in the literature. Schutte et al. (2012) reported worse function based on Child–Pugh score when comparing patients with HCC based on ALD vs. viral etiology of HCC [22]. Similar results, yet with even greater differences in proportions of patients within each Child–Pugh stage among ALD-HCC, viral HCC and NAFLD-related HCC, were also reported by Hester et al. [19]. Another study that corroborates this finding was published by Bucci et al. (2015). Both Child–Pugh (6.7 vs. 6.3, $p < 0.001$) and MELD scores (11.7 vs. 10.4, $p < 0.001$) were significantly worse in the ALD group at the time of HCC diagnosis [23]. Poorer metabolic function at the time of HCC diagnosis was also described by Constentin et al. (2018) in the prospective CHANGH cohort study. Again, a significantly lower proportion of Child–Pugh A and a higher proportion of Child–Pugh C patients was observed in the ALD-HCC group compared to non-ALD (39.3% in ALD vs. 66.0% in non-ALD for Child–Pugh A and 21.2% vs. 10.3% for Child–Pugh C [20]. Only a study by Trevisani et al. (2007) did not find a difference in metabolic liver function at the time of HCC diagnosis. In this study, the proportions of patients within the Child–Pugh A stage were comparable in all subgroups (67.8% of patients with ALD-HCC vs. 69.7% with HCV-related HCC vs. 61.4% with HBV-related HCC vs. 65.2% with multiple causes of chronic liver disease, $p = 0.787$) [24].

Besides liver function, patients with ALD-HCC also had poorer overall functional status measured by ECOG scale. Only about 74% of patients with ALD-HCC were in the ECOG stage 0 or 1 compared to almost 86% of patients with HCC due to other etiolo-

gies. This has also been confirmed in the literature, however, the data are not uniform. Costetin et al. also reported a lower proportion of ALD-HCC patients within ECOG stage 0 or 1, compared to non-ALD-HCC patients. However, it is important to note that in our cohort patients had better performance status overall compared to in the French cohort [20]. In the other study, Bucci et al. did not find any difference in performance status; however, his control group included only HCV patients with HCC and not a mixed control group as in our study and the study by Costentin et al. As HCV patients are commonly substance abusers this may explain their performance status comparable to ALD patients [23].

Interestingly, we have also found differences in the first line treatment of HCC among different etiologies of CLD. Patients with ALD-HCC, even in the BCLC 0–A stage, significantly more commonly received only supportive treatment compared to the control group. We did not observe differences in administered treatment in other BCLC stages. Similar results have been reported from the ITA.LI.CA database by Bucci et al., where as much as 30.4% of ALD-HCC patients received palliative treatment irrespective of their BCLC stage, compared to only 19.8% of patients with HCV-HCC. Furthermore, in the French cohort, authors reported that significantly fewer patients with ALD-HCC received potentially curative treatment compared to the control group (16.3% vs. 27.1%) [20], and this was confirmed in yet another study by deLemos et al. [21]. The aforementioned authors commonly cite reasons such as delayed diagnosis due to lower rates of HCC screening due to poorer access to healthcare, decreased compliance or even prejudice against alcohol consumers; however, the data exploring the causes are lacking. In addition, curative treatment strategies can provide slightly different long-term outcomes. In our study, all subjects underwent radiofrequency ablation; however, some meta-analysis favors microwave ablation in long-term follow-up [25].

Moreover, there are many additional cofactors that can influence hepatocellular cancer development and survival. Published meta-analyses suggest that regular statin [26], aspirin [27] and metformin [28] can reduce the risk of hepatocellular cancer development. Some of these can even improve the overall survival of patients with diagnosed cancer [29]. Because of the small number of regular users included in this study, this parameter was not analyzed. Diabetes mellitus is another well-accepted risk factor for hepatocellular cancer development [2], and some studies also confirmed its negative influence on recurrence-free survival [30] and overall survival [31]; however, in our study the univariant analysis did not confirm these outcomes in both subgroups.

Despite differences in performance status and treatment, no significant difference in overall survival (8.1 months in patients with HCC based on ALD vs. 8.5 months in non-ALD patients, $p = 0.315$) or DFS/RFS (4.9 months vs. 5.7 months, $p = 0.528$) was observed in our study. Not all previously published data on ALD-HCC survival are consistent. In the ITA.LI.CA cohort, authors reported lower overall survival in ALD-HCC patients (32.4 months) than in HCV patients (40.6 months; $P = 0.002$). Notably, the reported overall survival is extremely high compared to both our data and to data from the literature. More intriguing is the fact that patients with ALD-HCC had identical survival to the HCV-HCC patients when only patients from regular HCC surveillance program were analyzed [23]. Decreased overall survival in ALD-HCC patients was also reported in the French cohort (9.7 months in non-alcoholic group vs 5.7 months in patients with ALD ($P = 0.0002$); however, this difference disappeared when patients were separated into BCLC stages [20]. The largest cohort of HCC patients from the US multicenter study also found that ALD-HCC patients had lower overall survival compared to non-ALD-HCC patients (1.07 vs. 1.41 years, $P < 0.001$); however, this analysis was not adjusted for differences in treatment or BCLC stage [21]. We suppose that better molecular class and tumor characteristics are offset by poorer liver function and functional status, thus the survival is worse than, or at best not different from patients with HCC due to other etiologies.

The hallmark of alcoholic liver damage is the inflammation, which also plays a significant role in cancerogenesis. Approximately 90% of all HCCs arise from persistent chronic inflammatory process due to viral infection, NASH, or regular alcohol consumption [32].

Furthermore, alcohol consumption leads to increased gut permeability and increased PAMP (pathogen-associated molecular pattern) levels, which suppress the immune response by increasing the number of tumor-associated macrophages with M2 phenotype and MSDC cells that suppress the CD8+ cytotoxic anti-tumor immune response. In addition, the marked deposition of neutrophils in the tumor tissue of alcoholic steatohepatitis facilitates the escape of tumor cells from the immune response. Additionally, a significant overproduction of IL 1 and IL 17 in patients with HCC has been described [33]. Changes in the tumor microenvironment are also reflected in the peripheral blood. All three inflammatory indices used herein have shown that they significantly influence the course of hepatocellular cancer. Our study supports the influence of systemic inflammation as well as direct (geno)toxicity on the prognosis of HCC patients, and also serves as further validation of the respective cut-offs of these indices. The main topics for discussion in the field are the optimal cut-off and the inclusion of these indices in prognostic models such as BCLC to guide personalized therapy in HCC patients, which is a goal of future studies in the field.

### 5. Conclusions

The present study is the first study from Slovakia to collect real data on the risk factors for hepatocellular carcinoma. Alcoholic liver disease is the most common cause of hepatocellular carcinoma in Slovakia, accounting for almost 50% of cases. Patients with ALD-related HCC more commonly had cirrhosis that was in more advanced stages and had poorer performance status. Patients with ALD-related HCC were more frequently offered palliative treatment only, even in the BCLC 0–A stage. We observed no difference in survival between ALD-related HCC patients and control group HCC patients. Systemic inflammatory indexes are strong predictors of long-term mortality in patients with HCC. However, future research is needed to incorporate these indices into currently used management algorithms such as BCLC.

**Author Contributions:** Conceptualization, D.Š., M.J and P.J.; Data curation, D.Š. and M.Ž.; Formal analysis, J.G. and M.J.; Investigation, D.Š., I.A., S.A.-S., R.B., M.M., M.R., Ľ.S. and M.Ž.; Methodology, D.Š. and J.G.; Supervision, S.D. and P.J.; Writing—original draft, D.Š. and M.J.; Writing—review & editing, S.D., I.A., S.A.-S., R.B, M.M., M.R. and Ľ.S. All authors have read and agreed to the published version of the manuscript.

**Funding:** This research received no external funding.

**Institutional Review Board Statement:** The study protocol is in accordance with the 1964 Helsinki declaration, its later amendments, and the principles of good clinical practice. The study protocol was approved by The Ethics Committee of East Slovakia Oncological Institute, on 27 May 2021 (approval code, EK/2/05/2021).

**Informed Consent Statement:** The committee waived the need for the specific patients' informed consent due to the retrospective nature of the data collection (data already existed and were not a result of a research activity) and the usage of anonymous data only.

**Data Availability Statement:** Data sharing not applicable.

**Conflicts of Interest:** The authors declare no conflict of interest.

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
