# Peer review of "Alcoholic Liver Disease-Related Hepatocellular Carcinoma: Characteristics and Comparison to General Slovak Hepatocellular Cancer Population"

_curroncol, doi:10.3390/curroncol30030271_

Round 1

Reviewer 1 Report

Thank you very much for submitting your manuscript to the journal “Current Oncology”. Your manuscript is a well conducted retrospective multicenter study analyzing the characteristics of Hepatocellular carcinoma (HCC) in the Slovak population. It shows a high prevalence of HCCs caused by alcohol in Slovakia. Furthermore, it deals with the characteristics of HCCs caused by alcoholic liver disease. Attached you will find some issues regarding your manuscript.

Minor Issues:

1.     Line 30: Please introduce your conclusion.

2.     Line 44: “which may progress to”

3.     Line 90: “liver”

4.     Line 93: “)”

5.     Line 316: Please discuss potential impact of patient compliance and/or social status.

6.     Line 355: As well as direct (geno)toxicity

Author Response

Dear reviewer,

Thank you for your valuable input.

We have revised the manuscript according to your suggestions.

Best regards

Dr. Martin Janicko, PhD.

Minor Issues:

  1. Line 30: Please introduce your conclusion.
    Response: “added in conclusion”
  2. Line 44: “which may progress to”
    corrected
  3. Line 90: “liver”
    corrected
  4. Line 93: “)”
    corrected
  5. Line 316: Please discuss potential impact of patient compliance and/or social status.
    added: Aforementioned author commonly cite reasons such as delayed diagnosis due to lower rates of HCC screening due to worse access to healthcare, decreased compliance or even prejudice against alcohol consumers, however the data exploring the causes are lacking.
  6. Line 355: As well as direct (geno)toxicity
    added

Reviewer 2 Report

Dominik Šafčák et al. describe the incidence of Alcoholic liver disease related Hepatocellular carcinoma in Slovak Hepatocellular 

cancer population.

looks nice work impressive, but there are missing data in the introduction and discussion regarding the treatments of HCC

Author Response

Dear reviewer,

Thank you for your valuable input.

We have revised the manuscript according to your suggestions.

Best regards

Dr. Martin Janicko, PhD.

Rev 2

looks nice work impressive, but there are missing data in the introduction and discussion regarding the treatments of HCC.

We tried to make introduction as concise as possible, if the reviewer thinks it will benefit from some other information we will gladly include it.

discussion regarding the treatments of HCC

In line with reviewer 1 we have expanded the discussion about the differences in the treatment of the patients in our cohort. Full paragraph now states

“Interestingly, we have also found differences in the first line treatment of HCC among different etiologies of CLD. Patients with ALD HCC even in BCLC 0-A stage received significantly more commonly only supportive treatment compared to control group. We did not observe differences in administered treatment in other BCLC stages. Similar results have been reported from ITA.LI.CA database by Bucci et al. where as much as 30.4% of  ALD HCC patients received palliative treatment irrespective of BCLC stage, compared to only 19.8% of patients with HCV HCC. Also in the French cohort authors reported that significantly fewer patients with ALD HCC received potentially curative treatment compared to control group (16.3% vs. 27.1%) [20] and this was confirmed in yet another study by deLemos et al. [21]. Aforementioned authors commonly cite reasons such as delayed diagnosis due to lower rates of HCC screening due to worse access to healthcare, decreased compliance or even prejudice against alcohol consumers, however the data exploring the causes are lacking.”

Reviewer 3 Report

Very interesting paper. I have the following comments:

1) English grammar should be improved. I recommend the authors to have their manuscript revised by a native speaker.

2) How were the two cohorts matched? Maybe a propensity score matching would have been more suitable to this study

3) The authors should report the drugs assumed by the patients at baseline with particular regard to specific drugs that may influence the incidence and the clinical course of HCC, for instance statins (on this regard cite the recent meta-analysis PMID: 32260179)

4) Were the authors treated only with RFA or also with other ablative treatments such as MWA? On this regard comment that there are also other potential therapeutic options citing the recent meta-analysis PMID: 33339274 

5) Report how many patients were diabetic at baseline as this parameter may influence the outcomes

Author Response

Dear reviewer,

Thank you for your valuable input.

We have revised the manuscript according to your suggestions.

Best regards

Dr. Martin Janicko, PhD.

Rev. 3

1) English grammar should be improved. I recommend the authors to have their manuscript revised by a native speaker.
            English language has been revised.

2) How were the two cohorts matched? Maybe a propensity score matching would have been more suitable to this study.

            Since this was a cohort study that included all consecutive patients the risk of selection bias is non-existent (except minor proportion of patients with missing data). Since we did not select the „case“ group, there was no need to select „control“ group as well.

3) The authors should report the drugs assumed by the patients at baseline with particular regard to specific drugs that may influence the incidence and the clinical course of HCC, for instance statins (on this regard cite the recent meta-analysis PMID: 32260179)

            Only data on statin use were available, therefore this information has been added to table 1 + a brief paragraph in the discussion.
            See text : Results „The presence of Diabetes mellitus did not have an association with mortality, use of statins was infrequent, thus not included in the analysis.“
            Discussion: „Moreover, there are many additional cofunders, that can influence hepatocellular cancer development and survival. Published meta-analysis suggest, that regular statin [28]  aspirin [29].) and metformin [30] can reduce risk of hepatocellular cancer develop-ment. Some of them can even improve overall survival of patients with diagnosed can-cer[31]. Because of the small number of regular users included in this study, this parame-ter was not analyzed.“

4) Were the authors treated only with RFA or also with other ablative treatments such as MWA? On this regard comment that there are also other potential therapeutic options citing the recent meta-analysis PMID: 33339274 

            Out of possible ablation therapies only RFA was available in Slvoakia at the time period. However we have briefly mentioned the possibility of microwave ablation in tzhe discussion.

            In addition, curative treatment strategies can provide slightly different long-term outcomes. In our study all subjects underwent radiofrequency ablation, however some meta-analysis favors microwave ablation in long-term follow-up [27].

5) Report how many patients were diabetic at baseline as this parameter may influence the outcomes

            We have added the proportions of diabetics in table 1, along with a sentence in the results „The presence of Diabetes mellitus did not have an association with mortality, use of statins was infrequent, thus not included in the analysis.“ And a paragraph in the discussion „Diabetes mellitus is another well accepted risk factor for hepatocel-lular cancer development [2] and some studies also confirmed its negative influence on recurrence-free survival [32]  and overall survival[33], however in our study the univari-ant analysis did not confirmed this outcomes in both subgroups.“

Associated references were added to the bibliography.

Round 2

Reviewer 3 Report

The revised version of the manuscript is OK. Thank you!